# Who Leaked the Model? Tracking IP Infringers in Accountable Federated Learning

## Abstract

Federated learning (FL) emerges as an effective collaborative learning framework to coordinate data and computation resources from massive and distributed clients in training. Such collaboration results in non-trivial intellectual property (IP) represented by the model parameters that should be protected and shared by the whole party rather than an individual user. Meanwhile, the distributed nature of FL endorses a malicious client the convenience to compromise IP through illegal model leakage to unauthorized third parties. To block such IP leakage, it is essential to make the IP identifiable in the shared model and locate the anonymous infringer who first leaks it. The collective challenges call for *accountable federated learning*, which requires verifiable ownership of the model and is capable of revealing the infringer's identity upon leakage. In this paper, we propose Decodable Unique Watermarking (DUW) for complying with the requirements of accountable FL. Specifically, before a global model is sent to a client in an FL round, DUW encodes a client-unique key into the model by leveraging a backdoor-based watermark injection. To identify the infringer of a leaked model, DUW examines the model and checks if the triggers can be decoded as the corresponding keys. Extensive empirical results show that DUW is highly effective and robust, achieving over 99% watermark success rate for Digits, CIFAR-10, and CIFAR-100 datasets under heterogeneous FL settings, and identifying the IP infringer with 100% accuracy even after common watermark removal attempts.

## 1 Introduction

Federated learning (FL) (Konečnỳ et al., 2015) has been widely explored as a distributed learning paradigm to enable remote clients to collaboratively learn a central model without sharing their raw data, effectively leveraging the massive and diverse data available in clients for learning and protecting the data confidentiality. The learning process of FL models typically requires the coordination of significant computing resources from a multitude of clients to curate the valuable information in the client's data, and the FL models usually have improved performance than isolated learning and thus high commercial value. Recently, the risk of leaking such high-value models has drawn the attention of the public. One notable example is the leakage of the foundation model from Meta (Vincent, 2023) by users who gained the restricted distribution of models. The leakage through restricted distribution could be even more severe in FL which allows all participating clients to gain access to the valued model. For each iterative communication round, a central server consolidates models from various client devices, forming a global or central model. This model is then disseminated back to the clients for the next update, and therefore the malicious clients have full access to the global models. As such, effectively protecting the global models in FL is a grand challenge.

Watermarking techniques (Adi et al., 2018; Chen et al., 2021; Darvish Rouhani et al., 2019; Fan et al., 2019; Uchida et al., 2017; Zhang et al., 2018) are recently introduced to verify the IP ownership of models. Among them, backdoor-based watermarking shows strong applicability because of its model-agnostic nature, which repurposes the backdoor attacks of deep models and uses special-purposed data (trigger set) to insert hidden patterns in the model to produce undesired outputs given inputs with triggers (Zhang et al., 2018; Le Merrer et al., 2020; Goldblum et al., 2022; Li et al., 2022). A typical backdoor-based watermarking operates as follows: The model owner first generates a trigger set consisting of samples paired with pre-defined target labels. The owner then embeds the watermark into the model by fine-tuning the model with the trigger set and the original training

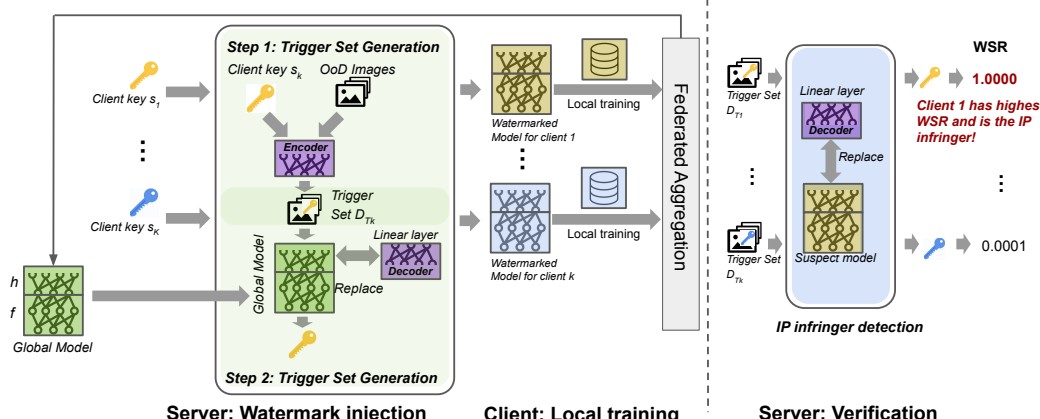

Figure 1: The proposed Decodable Unique Watermarking (DUW) for watermark injection and verification. During watermark injection, the server first uses client-unique keys and an OoD dataset as the input for the pre-trained encoder to generate trigger sets. When the server implants the watermark based on the objective function $J'(\theta_k)$ (Eq. (5)), a decoder is utilized to replace the classifier head. During verification, the suspect model is tested on all the trigger sets, and the client that leaked the model is identified as the one that achieves the highest WSR (Eq. (3)) in trigger sets.

samples. To establish the ownership of the model, one evaluates the accuracy of the suspect model using the trigger set. The mechanism safeguards the assumption that only the watermarked model would perform exceptionally well on the unique trigger set. If the model's accuracy on the trigger set surpasses a significant threshold, the model likely belongs to the owner.

Conventional backdoor-based watermarking, however, does not apply to FL settings because of the required access to the training data to maintain model utility. To address the challenge, Tekgul et al. (2021) proposed WAFFLE, which utilized only random noise and class-consistent patterns to embed a backdoor-based watermark into the FL model. However, since WAFFLE injected a unified watermark for all the clients, it cannot solve another critical question: *Who is the IP infringer among the FL clients?* Based on WAFFLE, Shao et al. (2022) introduced a two-step method FedTracker to verify the ownership of the model with the central watermark from WAFFLE, and track the malicious clients in FL by embedding unique local fingerprints into local models. However, the local fingerprint in Shao et al. (2022) is a parameter-based method, which is not applicable for many practical scenarios, where many re-sale models are reluctant to expose their parameters, and the two-step verification is redundant. Therefore, how to spend the least effort on changing the model while verifying and tracking the IP infringers using the same watermark in FL remains to be a challenging problem.

The aforementioned challenges call for a holistic solution towards *accountable federated learning*, which is characterized by the following essential requirements: R1) Accurate IP tracking: Each client has a unique ID to trace back. IP tracking should be confident to identify one and only one client. R2) Confident verification: The ownership verification should be confident. R3) Model utility: The watermark injected should have minimal impact on standard FL accuracy. R4) Robustness: The watermark should be robust and resilient against various watermark removal attacks. In this paper, we propose a practical watermarking framework for FL called Decodable Unique Watermarking (DUW) to comply with these requirements. Specifically, we first generate unique trigger sets for each client by using a pre-trained encoder (Li et al., 2021c) to embed client-wise unique keys to one randomly chosen out-of-distribution (OoD) dataset. During each communication round, the server watermarks the aggregated global model using the client-wise trigger sets before dispatching the model. A decoder replaces the classifier head in the FL model during injection so that we can decode the model output to the client-wise keys. We propose a regularized watermark injection optimization process to preserve the model's utility. During verification, the suspect model is tested on the trigger sets of all the clients, and the client that achieves the highest watermark success rate (WSR) is considered to be the IP infringer. The framework of method is shown in Fig. 1.

The contributions of our work can be summarized in three folds:
• We make the FL model leakage from anonymity to accountability by injecting DUW. DUW enables ownership verification and leakage tracing at the same time without access to model parameters during verification.

• With utility preserved, both the ownership verification and IP tracking of our DUW are not only accurate but also confident without collisions.
• Our DUW is robust against existing watermarking removal attacks, including fine-tuning, pruning, model extraction, and parameter perturbation.

## 2 RELATED WORK AND BACKGROUND

**Federated learning (FL)** is a distributed learning framework that enables massive and remote clients to collaboratively train a high-quality central model (Konečný et al., 2016). This paper targets the cross-silo FL with at most hundreds of clients (Marfoq et al., 2020). In the cross-silo setting, each client is an institute, like a hospital or a bank. It is widely adopted in practical scenario (Bagdasaryan et al., 2020; T Dinh et al., 2020; Zhu et al., 2021; Tekgul et al., 2021). FedAvg (McMahan et al., 2017) is one of the representative methods for FL, which averages local models during aggregation. This work is based on the FedAvg. Suppose we have $K$ clients, and our FL model $M$ used for standard training consists of two components, including a feature extractor $f : \mathcal{X} \to \mathcal{Z}$ governed by $\theta^f$, and a classifier $h : \mathcal{Z} \to \mathcal{Y}$ governed by $\theta^h$, where $\mathcal{Z}$ is the latent feature space. The collective model parameter is $\theta = (\theta^h, \theta^f)$. The objective for a client's local training is:

$$J_k(\theta) := \frac{1}{|\mathcal{D}_k|} \sum_{(x,y) \in \mathcal{D}_k} \ell(h(f(x; \theta^f); \theta^h), y), \tag{1}$$

where $\mathcal{D}_k$ is the local dataset for client $k$, and $\ell$ is the cross-entropy loss. The overall objective function of FL is thus given by $\min_{\theta} \frac{1}{K} \sum_{k=1}^{K} J_k(\theta)$.

**DNN watermarking** can be categorized into two main streams: parameter-based watermarking and backdoor-based watermarking.

*Parameter-based watermarking* approaches (Darvish Rouhani et al., 2019; Uchida et al., 2017; Kuribayashi et al., 2021; Mehta et al., 2022) embed a bit string as the watermark into the parameter space of the model. The ownership of the model can be verified by comparing the watermark extracted from the parameter space of the suspect model and the owner model. Shao et al. (2022) proposed a parameter-based watermarking method for FL called FedTracker. It inserts a unique parameter-based watermark into the models of each client to verify the ownership. However, all parameter-based watermarking requires an inspection of the parameters of the suspect models, which is not applicable enough for many re-sale models.

*Backdoor-based watermarking* (Zhang et al., 2018; Le Merrer et al., 2020; Goldblum et al., 2022; Li et al., 2022) does not require access to model parameters during verification. The watermark is embedded by fine-tuning the model with a trigger set $\mathcal{D}_T$ and clean dataset $\mathcal{D}$. Pre-defined target label $t$ is assigned to $\mathcal{D}_T$. The objective for the backdoor-based watermarking is formulated as:

$$J(\theta) := \frac{1}{|\mathcal{D}|} \sum_{(x,y) \in \mathcal{D}} \ell(h(f(x; \theta^f); \theta^h), y) + \frac{1}{|\mathcal{D}_T|} \sum_{(x,t) \in \mathcal{D}_T} \ell(h(f(x; \theta^f); \theta^h), t), \tag{2}$$

Upon verification, we verify the suspect model $M_s$ on the trigger set $\mathcal{D}_T$. If the accuracy of the trigger set is larger than a certain threshold $\sigma$, the ownership of the model can be established. We formally define the ownership verification of the backdoor-based model as follows:

**Definition 2.1** (Ownership verification). We define watermark success rate (WSR) as the accuracy on the trigger set $\mathcal{D}_T$:

$$\text{WSR} = Acc(M_s, \mathcal{D}_T). \tag{3}$$

If WSR $> \sigma$, the ownership of the model is established.

WAFFLE (Tekgul et al., 2021) is the first FL backdoor-based watermarking, which utilized random noise and class-consistent patterns to embed a backdoor-based watermark into the FL model. However, WAFFLE can only verify the ownership of the model, yet it cannot track the specific IP infringers.

## 3 METHOD

Watermarking has shown to be a feasible solution for IP verification, and the major goal of this work is to seek a powerful extension for traceable IP verification for accountable FL that can accurately identify the infringers among a scalable number of clients. A straightforward solution is injecting

different watermarks for different clients. However, increasing the number of watermarks could lower the model's utility as measured by the standard accuracy due to increased forged knowledge (Tang et al., 2020) (R3). Meanwhile, maintaining multiple watermarks could be less robust to watermark removal because of the inconsistency between injections (R4). Accurate IP tracking (R1) is one unique requirement we seek to identify the infringer's identity as compared with traditional watermarking in central training. The greatest challenge in satisfying R1 is addressing the watermark *collisions* between different clients. A watermark collision is when the suspect model produces similar watermark responses on different individual verification datasets in FL systems. Formally:

**Definition 3.1** (Watermark collision). During verification in Definition 2.1, we test the suspect model $M_s$ on all the verification datasets $\mathcal{D}_T = \{\mathcal{D}_{T_1}, \ldots, \mathcal{D}_{T_k}, \ldots, \mathcal{D}_{T_K}\}$ of all the clients to identify the malicious client, and WSR for the $k$-th verification datasets is defined as $\text{WSR}_k$. If we have multiple clients $k$ satisfying $\text{WSR}_k = Acc(M_s, \mathcal{D}_{T_k}) > \sigma$, the ownership of suspect model $M_s$ can be claimed for more than one client, then the watermark collisions happen between clients.

### 3.1 PITFALLS FOR WATERMARK COLLISION

To avoid watermark collision, one straightforward solution is to simply design different trigger sets for different clients. However, this strategy may easily lead to the watermark-collision pitfall. We use traditional backdoor-based watermarking by adding arbitrary badnet (Gu et al., 2019) triggers using random noise or 0-1 coding trigger for each client as examples to demonstrate this pitfall. We conduct the experiments on CIFAR-10 with 100 clients, during 4 injection rounds, at least $89\%$ and $87\%$ of the clients have watermark collisions for two kinds of triggers, respectively.

To analyze why these backdoor-based watermarkings lead us into the trap, we list all the clients with watermark collisions for one trial, and define the client_ID with the highest WSR as the predicted client_ID. We found that $87.5\%$ of the predicted client_ID share the same target label as the ground truth client, and for the rest $12.5\%$ clients, both the trigger pattern and target label are different. Based on the results, we summarize two possible reasons: 1) The same target labels will easily lead to the watermark collision. 2) The trigger pattern differences between clients are quite subtle, so the differences between the watermarked models for different clients are hard to detect. Thus, in order to avoid this pitfall, we have to ensure the uniqueness of both the triggers and target labels between different clients. More experiment settings and results for pitfalls can be referred to Appendix B.1.

### 3.2 DECODABLE UNIQUE WATERMARKING

In this section, we propose the Decodable Unique Watermark (DUW) that can simultaneously address the four requirements of accountable FL summarized in Section 1: R1 (accurate IP tracking), R2 (confident verification), R3 (model utility), R4 (robustness). In DUW, all the watermarking is conducted on the server side, so no computational overhead is introduced to clients. Before broadcasting the global model to each local client, the server will inject a unique watermark for each client. The watermark is unknown to clients but known to the server (see Fig. 1 server watermark injection). Our DUW consists of the following two steps for encoding and decoding the client-unique keys.

**Step 1: Client-unique trigger encoding.** Due to the data confidentiality of FL, the server has no access to any data from any of the clients. Therefore for watermark injection, the server needs to collect or synthesize some OoD data for trigger set generation. The performance of the watermark is not sensitive to the choice of the OoD datasets.

To accurately track the malicious client, we have to distinguish between watermarks for different clients. High similarity between trigger sets of different clients is likely to cause watermark collisions among the clients (see Section 3.1), which makes it difficult to identify which client leaked the model.

To solve this problem, we propose to use a pre-trained encoder $E : \mathcal{X} \rightarrow \mathcal{X}$ governed by $\theta_E$ from Li et al. (2021c) to generate unique trigger sets for each client. This backdoor-based method provides a successful injection of watermarks with close to $100\%$ WSR, which ensures the confident verification (R2). We design a unique key corresponding to each client ID as a one-hot binary string to differentiate clients. For instance, for the $k$-th client, the $k$-th entry of the key string $s_k$ is 1, and the other entries are 0. We set the length of the key as $d$, where $d \geq K$. For each client, the key can then be embedded into the sample-wise triggers of the OoD samples by feeding the unique key and OoD data to the pre-trained encoder. The output of the encoder makes up the trigger sets. The trigger

set for the $k$-th client is defined in $\mathcal{D}_{T_k} = \{(x', t_k) | x' \sim E_{x \in \mathcal{D}_{OoD}}(x, s_k; \theta_E)\}$, where $\mathcal{D}_{OoD}$ is a randomly chosen OoD dataset, and $t_k$ is the target label for client $k$. To this end, different trigger sets for different clients will differ by their unique keys, and watermark collision can be alleviated (R1). Note that our trigger sets will be the same as verification datasets.

**Step 2: Client-unique target label by decoding triggers to client keys.** The main intuition is that the same target label of the trigger sets may still lead to watermark collisions even if the keys are different (see Section 3.1). Thus, we propose to project the output dimension of the original model $M$ to a higher dimension, larger than the client number $K$, to allow each client to have a unique target label. To achieve this goal, we first set the target label $t_k$ in the trigger set $\mathcal{D}_{T_k}$ to be the same as the input key $s_k$ corresponding to each client, and then use a decoder $D : \mathcal{Z} \to \mathcal{Y}$ parameterized by $\theta_D$ to replace the classifier $h$ in the FL training model $M$. The decoder $D$ only has one linear layer, whose input dimension is the same as the input dimension of $h$, and its output dimension is the length of the key. To avoid watermark collision between clients induced by the target label, we make the decoder weights orthogonal with each other during the random initialization so that the watermark injection tasks for each client can be independent (R1). The weights of the decoder are frozen once initialized to preserve the independence of different watermark injection tasks for different clients. Suppose $\theta_k = (\theta_k^f, \theta_k^h)$ is the parameter which will be broadcast for client $k$, we formulate the injection optimization as:

$$\min_{\theta_k^f} J(\theta_k^f) := \frac{1}{|\mathcal{D}_{T_k}|} \sum_{(x', s_k) \in \mathcal{D}_{T_k}} \ell(D(f(x'; \theta_k^f); \theta_D), s_k), \tag{4}$$

The classifier $h$ will be plugged back into the model before the server broadcasts the watermarked models to clients. Compared with traditional backdoor-based watermarking (Eq. (2)), our watermark injection requires no client training samples, which ensures the data confidentiality of FL.

**Robustness.** Our framework also brings in robustness against fine-tuning-based watermark removal (R4). The main intuition is that replacing classifier $h$ with decoder $D$ also differs the watermark injection task space from the original classification task space. Since the malicious clients have no access to the decoder and can only conduct attacks on model $M$, the attacks have more impact on the classification task instead of our watermark injection task, which makes our decodable watermark more resilient against watermark removal attacks.

## 3.3 INJECTION OPTIMIZATION WITH PRESERVED UTILITY

While increasing the size of the client number, watermark injection in the OoD region may lead to a significant drop in the standard FL accuracy (R3) because of the overload of irrelevant knowledge. An ideal solution is to bundle the injection with training in-distribution (ID) data, which however is impractical for a data-free server. Meanwhile, lacking ID data to maintain the standard task accuracy, the distinct information between the increasing watermark sets and the task sets could cause the fade-out of the task knowledge. We attribute such knowledge vanishing to the divergence in the parameter space between the watermarked and the original models. Thus, we propose to augment the injection objective Eq. (4) with a $l_2$ regularization on the parameters:

$$\min_{\theta_k} J'(\theta_k^f) := J(\theta_k^f) + \frac{\beta}{2} \|\theta_k^f - \theta_g^f\|^2, \tag{5}$$

where $\theta_g^f$ is the original parameter of the global model. The regularization term of Eq. (5) is used to restrict the distance between the watermarked model and the non-watermarked one so that the utility of the model can be better preserved (R3). Our proposed DUW is summarized in Algorithm 1.

## 3.4 VERIFICATION

During verification, we not only verify whether the suspect model $M_s = (f_s, h_s)$ is a copy of our model $M$, but also track who is the leaker among all the clients by examining if the triggers can be decoded as the corresponding keys. To achieve this goal, we first use our decoder $D$ to replace the classifier $h_s$ in the suspect model $M_s$, then the suspect model can be restructured as $M_s = (f_s, D)$. According to Definition 3.1, we test the suspect model $M_s$ on all the verification datasets $\mathcal{D}_T = \{\mathcal{D}_{T_1}, \ldots, \mathcal{D}_{T_k}, \ldots, \mathcal{D}_{T_K}\}$ of all the clients to track the malicious clients, and report $\text{WSR}_k$ on the $k$-th verification datasets correspondingly. The client whose verification dataset achieves

the highest WSR leaked the model (see Fig. 1 server verification). The tracking mechanism can be defined as $Track(M_s, \mathcal{D}_T) = \arg\max_k \text{WSR}_k$.

Suppose the ground truth malicious client is $k_m$. If $\text{WSR}_{k_m} > \sigma$, and $\text{WSR}_k$ for other verification datasets is smaller than $\sigma$, then the ownership of the model can be verified, and no watermark collision happens. If $Track(M_s, \mathcal{D}_T) = k_m$, then the malicious client is identified correctly.

---

**Algorithm 1** Injection of Decodable Unique Watermarking (DUW)

---

1: **Input:** Clients datasets $\{\mathcal{D}_k\}_{k=1}^K$, OoD dataset $\mathcal{D}_{\text{OoD}}$, secret key $\{s_k\}_{k=1}^K$, pre-trained encoder $E$, pre-defined decoder $D$, global parameters $\theta_g$, local parameters $\{\theta_k\}_{k=1}^K$, learning rate $\alpha, \beta$, local training steps $T$, watermark injection steps $T_w$.
2: **Step 1: Client-unique trigger encoding.**
3: **for** $k = 1, \ldots, K$ **do**
4:     Generate trigger set for client $k$: $\mathcal{D}_{T_k} = \{(x', s_k) | x' \sim E_{x \in \mathcal{D}_{OoD}}(x, s_k; \theta_E)\}$
5: **end for**
6: **Step 2: Decoding triggers to client keys.**
7: **repeat**
8:     Server selects active clients $\mathcal{A}$ uniformly at random
9:     **for** all client $k \in \mathcal{A}$ **do**
10:         Server initializes watermarked model for client $k$ as: $\theta_k \leftarrow \theta_g$.
11:         **for** $t = 1, \ldots, T_w$ **do**
12:             Server replaces model classifier $h$ with decoder $D$.
13:             Server injects watermark to model using trigger set $\mathcal{D}_{T_k}$, and update $\theta_k^f$ as:
                $\theta_k^f \leftarrow \theta_k^f - \beta \nabla_{\theta_k^f} J'(\theta_k^f)$.    ▷ Optimize Eq. (5)
14:         **end for**
15:         Server broadcasts $\theta_k$ to the corresponding client $k$.
16:         **for** $t = 1, \ldots, T$ **do**
17:             Client local training using local set $\mathcal{D}_k$: $\theta_k \leftarrow \theta_k - \alpha \nabla_{\theta_k} J_k(\theta_k)$.    ▷ Optimize Eq. (1)
18:         **end for**
19:         Client $k$ sends $\boldsymbol{\theta}_k$ back to the server.
20:     **end for**
21:     Server updates $\theta_g \leftarrow \frac{1}{|\mathcal{A}|} \sum_{k \in \mathcal{A}} \theta_k$.
22: **until** training stop

---

## 4 EXPERIMENTS

In this section, we empirically show how our proposed DUW can fulfill the requirements (R1-R4) for tracking infringers as described in Section 1.

**Datasets.** To simulate *class non-iid* FL setting, we use CIFAR-10, CIFAR-100 (Krizhevsky et al., 2009), which contain $32 \times 32$ images with 10 and 100 classes, respectively. CIFAR-10 data is uniformly split into 100 clients, and 3 random classes are assigned to each client. CIFAR-100 data is split into 100 clients with Dirichlet distribution. For CIFAR-10 and CIFAR-100, the OoD dataset we used for OoD injection is a subset of ImageNet-DS (Chrabaszcz et al., 2017) with randomly chosen 500 samples downsampled to $32 \times 32$. To simulate the *feature non-iid* FL setting, a multi-domain FL benchmark, Digits (Li et al., 2020; Hong et al., 2022) is adopted. The dataset is composed of $28 \times 28$ images for recognizing 10 digit classes, which was widely used in the community (Caldas et al., 2018; McMahan et al., 2017). The Digits includes five different domains: MNIST (LeCun et al., 1998), SVHN (Netzer et al., 2011), USPS (Hull, 1994), SynthDigits (Ganin & Lempitsky, 2015), and MNIST-M (Ganin & Lempitsky, 2015). We leave out USPS as the OoD dataset for watermark injection (500 samples are chosen) and use the rest four domains for the standard FL training. Each domain of digits is split into 10 different clients, thus, 40 clients will participate in the FL training.

**Training setup.** A preactivated ResNet (PreResNet18) (He et al., 2016) is used for CIFAR-10, a preactivated ResNet (PreResNet50) (He et al., 2016) is used for CIFAR-100, and a CNN defined in Li et al. (2021b) is used for Digits. For all three datasets, we leave out 10% of the training set as the validation dataset to select the best FL model. The total training round is 300 for CIFAR-10 and CIFAR-100, and 150 for Digits.

| Dataset | Acc | $\Delta$Acc | WSR | WSR_Gap | TAcc |
|---------|-----|-------------|-----|---------|------|
| Digits | 0.8855 | 0.0234 | 0.9909 | 0.9895 | 1.0000 |
| CIFAR-10 | 0.5583 | 0.0003 | 1.0000 | 0.9998 | 1.0000 |
| CIFAR-100 | 0.5745 | 0.0063 | 1.0000 | 0.9998 | 1.0000 |

Table 1: Benchmark results.

**Watermark injection.** The early training stage of FL is not worth protecting since the standard accuracy is very low, we start watermark injection at round 20 for CIFAR-10 and Digits, and at round 40 for CIFAR-100. The standard accuracy before our watermark injection is 85.20%, 40.23%, and 29.41% for Digits, CIFAR-10, and CIFAR-100, respectively.

**Evaluation metrics.** For watermark verification, we use watermark success rate (**WSR**) which is the accuracy of the trigger set for evaluation. To measure whether we track the malicious client (leaker) correctly, we define tracking accuracy (**TAcc**) as the rate of the clients we track correctly. To further evaluate the ability of our method for distinguishing between different watermarks for different clients, we also report the difference between the highest WSR and second best WSR as **WSR_Gap** to show the significance of verification and IP tracking. With a significant WSR_Gap, no watermark collision will happen. To evaluate the utility of the model, we report the standard FL accuracy (**Acc**) for each client's individual test sets, whose classes match their training sets. We also report the accuracy degradation ($\Delta$**Acc**) of the watermarked model compared with the non-watermarked one. Note that, to simulate the scenario where malicious clients leak their local model after local training, we test the average WSR, TAcc and WSR_Gap for the local model of each client instead of the global model. Acc and $\Delta$Acc are evaluated on the best FL model selected using the validation datasets.

## 4.1 IP TRACKING BENCHMARK

We evaluate our method using the IP tracking benchmark with various metrics as shown in Table 1. Our ownership verification is confident with all WSRs over 99% (R2). The model utility is also preserved with accuracy degradation 2.34%, 0.03%, and 0.63%, respectively for Digits, CIFAR-10 and CIFAR-100 (R3). TAcc for all benchmark datasets is 100% which indicates accurate IP tracking (R1). All WSR_Gap is over 98%, which means the WSRs for all other benign client's verification datasets are close to 0%. In this way, the malicious client can be tracked accurately with high confidence, no collisions will occur within our tracking mechanism (R1).

## 4.2 ROBUSTNESS

Malicious clients can conduct watermark removal attacks before leaking the FL model to make it harder for us to verify the model copyright, and track the IP infringers accurately. In this section, we show the robustness of the watermarks under various watermark removal attacks (R4). Specifically, we evaluate our method against 1) **fine-tuning** (Adi et al., 2018): Fine-tune the model using their own local data; 2) **pruning** (Liu et al., 2018): prune the model parameters that have the smallest absolute value according to a certain pruning rate, and then fine-tune the model on their local data; 3) **model extraction attack**: first query the victim model for the label of an auxiliary dataset, and then re-train the victim model on the annotated dataset. We take knockoff (Orekondy et al., 2019) as an example of the model extraction attack; 4) **parameter perturbations**: add random noise to local model parameters (Garg et al., 2020).

10 of the clients are selected as the malicious clients, and the metrics in this section are average values for 10 malicious clients. All the watermark removal attacks are conducted for 50 epochs with a learning rate $10^{-5}$. All the attacks are conducted for the local model of the last round.

**Robustness against fine-tuning attack.** We report the robustness of our proposed DUW against fine-tuning in Table 2. $\Delta$Acc and $\Delta$WSR in this table indicate the accuracy and WSR drop compared with accuracy and WSR before the attack. According to the results, after 50 epochs of fine-tuning, the attacker can only decrease the WSR by less than 1%, and the TAcc is even not affected. Fine-tuning with their limited local training samples can also cause a standard accuracy degradation. Fine-tuning can neither remove our watermark nor affect our IP tracking, even if sacrifices their standard accuracy.

**Robustness against pruning attack.** We investigate the effect of pruning in Fig. 2 by varying the pruning rate from 0 to 0.5. With the increase in the pruning ratio, both TAcc and WSR will not be

| Dataset | Acc | $\Delta$Acc | WSR | $\Delta$WSR | TAcc |
|---|---|---|---|---|---|
| Digits | 0.9712 | -0.0258 | 0.9924 | 0.0030 | 1.0000 |
| CIFAR-10 | 0.7933 | 0.1521 | 1.0000 | 0.0000 | 1.0000 |
| CIFAR-100 | 0.4580 | 0.0290 | 0.9930 | 0.0070 | 1.0000 |

Table 2: DUW is robust against fine-tuning.

| Dataset | Acc | $\Delta$Acc | WSR | $\Delta$WSR | TAcc |
|---|---|---|---|---|---|
| Digits | 0.8811 | 0.0643 | 0.9780 | 0.0174 | 1.0000 |
| CIFAR-10 | 0.5176 | 0.0010 | 0.6638 | 0.3362 | 1.0000 |
| CIFAR-100 | 0.4190 | 0.0680 | 0.8828 | 0.1172 | 1.0000 |

Table 3: DUW is robust against model extraction.

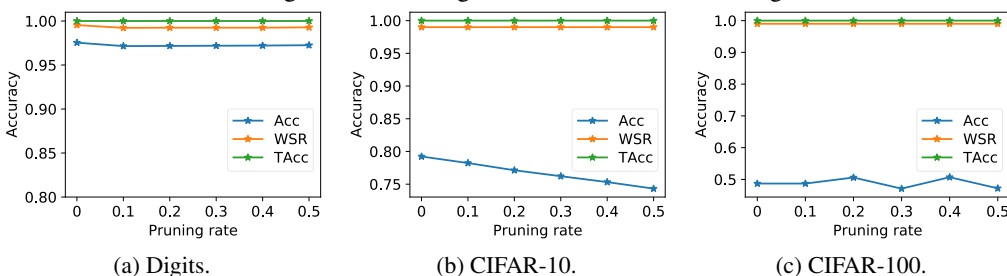

(a) Digits.     (b) CIFAR-10.     (c) CIFAR-100.

Figure 2: DUW is robust against pruning.

affected. For CIFAR-10, standard accuracy will drop $5\%$. Therefore, pruning is not an effective attack on our watermark, and it will even cause an accuracy degradation for the classification task.

**Robustness against model extraction attack.** To verify the robustness of our proposed DUW against model extraction attack, we take knockoff (Orekondy et al., 2019) as an example, and STL10 (Coates et al., 2011) cropped to the same size as the training data is used as the auxiliary dataset for this attack. According to the results for three benchmark datasets in Table 3, after knockoff attack, WSR for all three datasets is still over $65\%$, and our tracking mechanism is still not affected with TAcc remains to be $100\%$. Therefore, our DUW is resilient to model extraction attacks.

**Robustness against parameter perturbations attack.** Malicious clients can also add random noise to model parameters to remove watermarks, since Garg et al. (2020) found that backdoor-based watermarks are usually not resilient to parameter perturbations. Adding random noise to the local model parameters can also increase the chance of blurring the difference between different watermarked models. We enable each malicious client to blend Gaussian noise to the parameters of their local model, and set the parameter of the local model as $\theta_i = \theta_i + \theta_i * \alpha_{\text{noise}}$, where $\alpha_{\text{noise}} = \{10^{-5}, 10^{-4}, 10^{-3}, 10^{-2}, 10^{-1}\}$. We investigate the effect of parameter perturbation in Fig. 3. According to the results, when $\alpha_{\text{noise}}$ is smaller than $10^{-2}$, WSR, Acc, and TAcc will not be affected. When $\alpha_{\text{noise}} = 10^{-2}$, Acc will drop more than $10\%$, TAcc remains unchanged, and WSR is still over $90\%$. When $\alpha_{\text{noise}} = 10^{-1}$, Acc will drop to a random guess, thus, although the watermark has been removed, the model has no utility. Therefore, parameter perturbation is not an effective attack for removing our watermark and affecting our tracking mechanism.

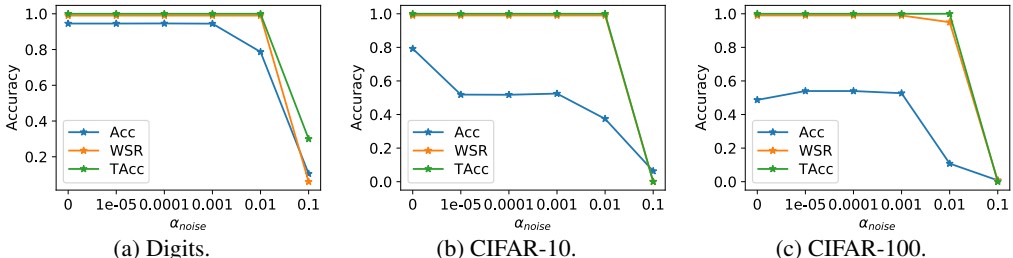

(a) Digits.     (b) CIFAR-10.     (c) CIFAR-100.

Figure 3: DUW is robust against parameter perturbation.

### 4.3 QUALITATIVE STUDY

**Effects of decoder.** To investigate the effects of the decoder on avoiding watermark collision, we compare the results of w/ and w/o decoder. When the decoder is removed, the task dimension of the watermark injection will be the same as the FL classification, thus, we also have to change the original target label (the same as the input key) to the FL classification task dimension. To achieve this goal, we set the target label of w/o decoder case as (client_ID % class_number). We report the results of w/ and w/o decoder on CIFAR-10 after 1 round of watermark injection at round 20 in Table 4. According to the results, when we have 100 clients in total, w/o decoder can only achieve a TAcc of $6\%$, while w/ decoder can increase TAcc to $100\%$. We also find that clients with the same

| Method | Acc | $\Delta$Acc | WSR | TAcc |
|---|---|---|---|---|
| w/ decoder | 0.3287 | 0.0736 | **0.8778** | **1.0000** |
| w/o decoder | 0.3235 | 0.0788 | 0.8099 | 0.0600 |

Table 4: Effects of decoder: the decoder can improve TAcc to avoid watermark collision. $\Delta$Acc in this table is the accuracy degradation compared with the previous round.

| Dataset | Acc | $\Delta$Acc | WSR | WSR_Gap | TAcc |
|---|---|---|---|---|---|
| USPS | 0.8855 | 0.0234 | 0.9909 | 0.9895 | 1.0000 |
| GTSRB | 0.8716 | 0.0373 | 0.9972 | 0.9962 | 1.0000 |
| Random noise | 0.9007 | 0.0082 | 0.8422 | 0.8143 | 1.0000 |
| Jigsaw | 0.9013 | 0.0076 | 0.8789 | 0.8601 | 1.0000 |

Table 5: Effects of different OoD datasets: a trade-off exists between Acc and WSR, given different selections of OoD datasets.

target label are more likely to conflict with each other, which makes those clients difficult to be identified, even if their trigger sets are different. Utilizing a decoder to increase the target label space to a dimension larger than the client number allows all the clients to have their own target label. In this way, watermark collision can be avoided. Besides, WSR of w/ decoder is also higher than w/o decoder after 1 round of injection. One possible reason is that we differ the watermark injection task from the original classification task using the decoder, thus, in this case, the watermark will be more easily injected compared with directly injected to the original FL classification task.

**Effects of $l_2$ regularization.** To show the effects of $l_2$ regularization in Eq. (5), we report the validation accuracy and WSR for 4 rounds of watermark injection on Digits with different values of the hyperparameter $\beta$ in Fig. 4a. Validation accuracy is the standard FL accuracy evaluated on a validation dataset for every round. We see that with the increase of $\beta$, higher validation accuracy can be achieved, but correspondingly, WSR drops from over 90% to only 35.65%. Larger $\beta$ increases the impact of $l_2$ norm, which decreases the model difference between the watermarked model and the non-watermarked one, so the validation accuracy will increase. At the same time, the updates during watermark injection also have much more restriction due to $l_2$ regularization, so the WSR drops to a low value. Accordingly, we select $\beta = 0.1$ for all our experiments, since $\beta = 0.1$ can increase validation accuracy by 6.88% compared with $\beta = 0$, while maintaining WSR over 90%.

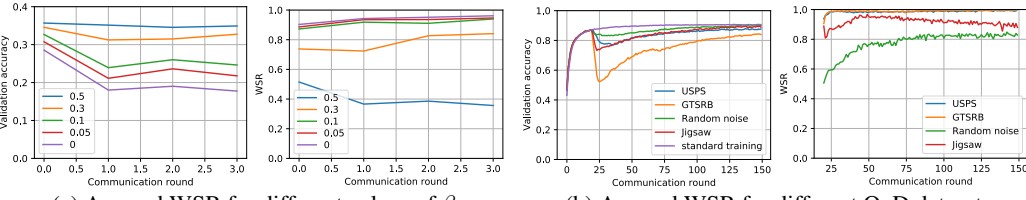

(a) Acc and WSR for different values of $\beta$.   (b) Acc and WSR for different OoD datasets.

Figure 4: Acc and WSR w.r.t. different communication rounds.

**Effects of different OoD datasets for watermark injection.** We investigate the effects of different OoD datasets including USPS (Hull, 1994), GTSRB (Stallkamp et al., 2012), random noise, and Jigsaw for watermark injection when the standard training data is Digits. All OoD images are cropped to the same size as the training images. A jigsaw image is generated from a small $4 \times 4$ random image, and then uses reflect padding mode from PyTorch to padding to the same size as the training images. The effect of these different OoD datasets is shown in Table 5 and Fig. 4b. We see that all OoD datasets can achieve 100% TAcc, suggesting the selection of OoD dataset will not affect the tracking of the malicious client. There is a trade-off between the Acc and WSR: higher WSR always leads to lower Acc. Random noise and jigsaw achieve high Acc, with accuracy degradation within 1%. These two noise OoD also have a faster recovery of the standard accuracy after the accuracy drop at the watermark injection round as shown in Fig. 4b, but the WSR of random noise and Jigsaw are lower than 90%. For two real OoD datasets USPS and GTSRB, the WSR quickly reaches over 99% after 1 communication round, but their accuracy degradation is larger than 2%.

## 5 CONCLUSION

In this paper, we target at accountable FL, and propose Decodable Unique Watermarking (DUW), that can verify the FL model's ownership and track the IP infringers in the FL system at the same time. Specifically, the server will embed a client-unique key into each client's local model before broadcasting. The IP infringer can be tracked according to the decoded keys from the suspect model. Extensive experimental results show the effectiveness of our method in accurate IP tracking, confident verification, model utility preserving, and robustness against various watermark removal attacks.

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
