## A    DISCUSSIONS

**Client-side watermarking VS server-side watermarking.** Client-side watermarking such as FedCIP (Liang & Wang, 2023), and Merkle-Sign (Li et al., 2021a) are used to claim the co-ownership of the model, yet we argue that client-side watermarking has some limitations, which makes it not applicable for IP tracking. For client-side watermarking, if one of the clients is the infringer to illegally distribute the model, the infringer will not reveal their own identity during the model verification process in order to avoid legal responsibility. Even if the ownership of the model can be claimed by their co-author, the real infringers cannot be tracked, since they remain anonymous. Using our server-side watermark, there is no such concern, the server can easily track the malicious client among all the clients.

**Complexity.** Clients will not experience additional computations as our DUW is carried out on the server side. The additional computation for the server is decided by the number of watermark injection steps $T_w$. We found that WSR could reach 99% just within $T_w = 10$ steps. Injection of one client-unique watermark takes around 1 second. The server can embed the watermark parallelly for all the clients. Since the watermarked model for each client is independent and has no sequence relationship with each other, there is no need to serialize it. Thus, the delay caused by the server is neglectable.

**Future works.** This paper makes the FL model leakage from anonymity to accountability by injecting client-unique watermarks. We recognize the most significant challenge for accountable FL is addressing watermark collision for accurate IP tracking (R1). We believe it is important to scale our method with more clients in the future. One plausible solution is increasing the dimension of the input of the encoder to allow more one-hot encoding target labels. Another solution is to use a hash function as the target label for different clients. In this way, the lower-dimensional encoder and decoder can accommodate more clients. For instance, an encoder with input dimension 10 can allow at most 1024 different clients. However, adopting hash functions as the target labels can increase the chance of watermark collision between clients, and more elegant strategies have to be developed to address this problem. As we focus on the collision, we leave the scalability for future work.

## B    SUPPLEMENTARY EXPERIMENTS

### B.1    COMPARISON WITH TRADITIONAL BACKDOOR-BASED WATERMARKS

We compare our proposed DUW with two traditional backdoor-based watermarks in Fig. 5. Due to the reason that if all the clients share the same trigger, watermark collision will definitely happen, we design different triggers for different clients. Specifically, we use traditional backdoor-based watermarking by adding arbitrary badnet triggers using random noise or 0-1 coding trigger for each client. To distinguish between different clients, for 0-1 trigger, following  Tang et al. (2020), we set 5 pixel values of the pattern into 0 and other 11 pixels into 1, different combinations of the pattern are randomly chosen for different clients. For random noise triggers, we generate different random noise triggers for different clients. The trigger size $4 \times 4$ and the injection is conducted for 4 rounds. The target label for each client is set as (client_ID % class_number). According to the results, traditional backdoor-based watermarks can only achieve a tracking accuracy lower than 13% (it will even be lower with the increase of the communication rounds), which is much lower than the 100% tracking accuracy we have achieved. Note that, the rate of clients with watermark collisions can be calculated as 1-TAcc.

To analyze the failure of the traditional backdoor-based watermarking, we give detailed prediction results for one trial on CIFAR10 for random noise trigger as an example. The client number is 100, so the client_ID is from 0-99, and the class number is 10. Here we provide a fine-grained analysis of the concerned 13% TAcc by looking into the last 10 clients. We list the client_ID and their corresponding predicted client_ID for clients 90-99 in Table 6. From the prediction results, 8 of 10 clients are tracked wrong. Among these 8 failure cases, 7 of the predicted client_ID (client 90, 91, 93, 94 95, 96, 98) share the same targets, and 1 of them (client 97) have both different triggers and different target labels. The two kinds of failures correspond to two different reasons respectively as we illustrated in Section 3.1.

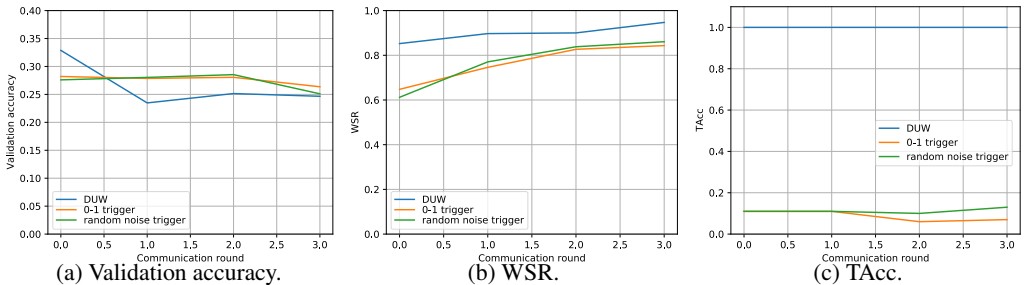

(a) Validation accuracy.  (b) WSR.  (c) TAcc.

Figure 5: Validation accuracy, WSR, and TAcc for proposed DUW and other two baselines on CIFAR-10 for 4 communication rounds.

| ground truth client_ID | predicted client_ID |
|:---:|:---:|
| 90 | 0 |
| 91 | 1 |
| 92 | 92 |
| 93 | 3 |
| 94 | 4 |
| 95 | 5 |
| 96 | 6 |
| 97 | 49 |
| 98 | 8 |
| 99 | 99 |

Table 6: Prediction results for random noise trigger for client 90-99.

## B.2 EXTENDED QUALITATIVE STUDY

**Visualization of unique trigger sets.** We show the visualization example for the original image, encoded image (image in trigger set), and residual image based on different OoD datasets in Fig. 6. We observe that for all four different OoD datasets, the original image and encoded image with our client keys are indistinguishable from the human eye. The difference between these two images can be observed in the residual image. Note that although the OoD datasets are different, the encoder that we used to generate the trigger sets is the same. According to Fig. 6, the encoder will generate sample-wise triggers for different images.

To investigate the difference between different clients' trigger sets based on the same OoD dataset, we show one example in the trigger set generated by the jigsaw image for two randomly picked clients in Fig. 7. The trigger sets are generated based on the same jigsaw dataset and differ by their embedded keys. According to Fig. 7, although the samples from different trigger sets do not look distinguishable according to the human inspection, the difference between keys decoded from the trigger sets can be distinguished by our model.

**Effects of the different numbers of samples in trigger sets.** We investigate how the size of the trigger set will affect our watermark injection and standard FL training in Fig. 8 by varying the number of samples in the trigger set from 50 to 500 for Digits training (USPS is used to generate the trigger set). Note that for all cases, TAcc always remains to be 100%. We observe that with only 50 samples in one trigger set, we can achieve an accuracy degradation around 2%, and with a WSR over 98%. When the number of samples increases to 300, WSR is over 99%. In general, the change in the number of samples in the trigger set has almost no effect on both standard accuracy and WSR. A small trigger set (such as 50) can achieve comparable results with a large trigger set. The advantages of a smaller trigger set include quicker trigger set generation, quicker watermark injection, quicker ownership verification, and quicker IP tracking. Besides, less effort can be made for OoD data synthesizing or collecting.

**Effects of different watermark injection rounds.** We conduct an ablation study to show the effect of the injection round of the watermark in Table 7. The results verify that injecting in earlier rounds will not affect standard accuracy, WSR, and TAcc. In our paper, we do not start our watermark injection

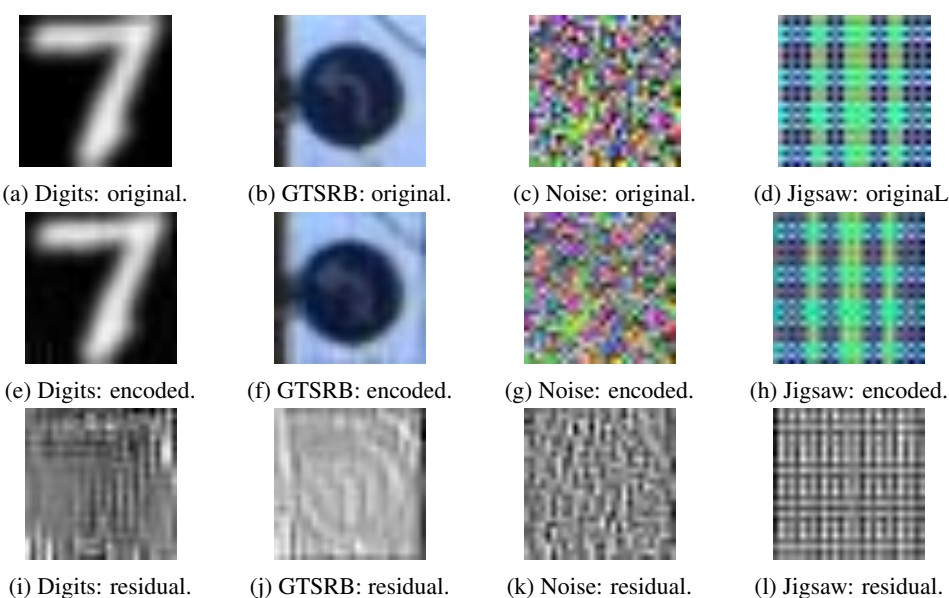

(a) Digits: original.  (b) GTSRB: original.  (c) Noise: original.  (d) Jigsaw: originaL.

(e) Digits: encoded.  (f) GTSRB: encoded.  (g) Noise: encoded.  (h) Jigsaw: encoded.

(i) Digits: residual.  (j) GTSRB: residual.  (k) Noise: residual.  (l) Jigsaw: residual.

Figure 6: Visualization of unique trigger set based on different OoD datasets.

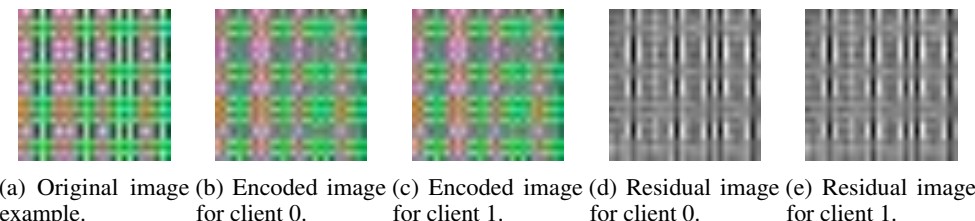

(a) Original image example.  (b) Encoded image for client 0.  (c) Encoded image for client 1.  (d) Residual image for client 0.  (e) Residual image for client 1.

Figure 7: Visualization of the unique trigger sets for two different clients. The difference between trigger sets cannot be observed according to human inspection, but after decoding, the difference between keys can be distinguished by our model.

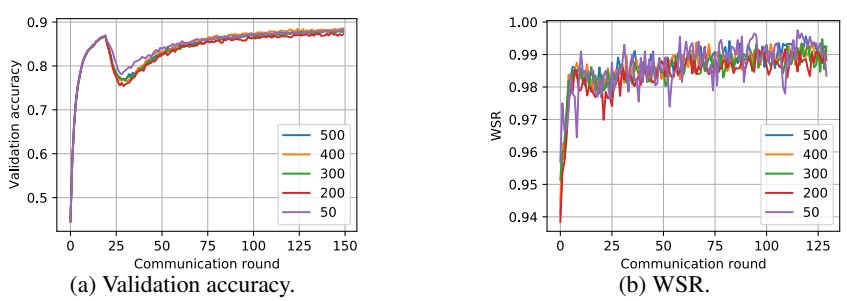

(a) Validation accuracy.  (b) WSR.

Figure 8: Effects of the different number of samples in trigger sets. 50 samples in one trigger set can achieve over 98% WSR.

at the very beginning of training since early-stage protection usually means more computational resources, so it is more valuable to focus on high-quality models rather than low-quality models.

| Inject round | Acc | $\triangle$Acc | WSR | WSR_Gap | TAcc |
|---|---|---|---|---|---|
| 5 | 0.8838 | 0.0251 | 0.9951 | 0.9948 | 1.0000 |
| 10 | 0.8811 | 0.0278 | 0.9946 | 0.9938 | 1.0000 |
| 20 | 0.8855 | 0.0234 | 0.9909 | 0.9895 | 1.0000 |

Table 7: Ablation study: results for watermark injection in different rounds on digits.

**Effects for different number of clients** We conduct an ablation study to show the effect of the number of clients in Table 8. According to the results, even with 600 clients, the WSR is still over 73% and the TAcc remains 100%. With more clients participating in FL, we can still track the malicious client correctly with high confidence.

| Number of clients | Acc | $\triangle$Acc | WSR | WSR_Gap | TAcc |
|---|---|---|---|---|---|
| 40 | 0.8855 | 0.0234 | 0.9909 | 0.9895 | 1.0000 |
| 400 | 0.8597 | -0.0332 | 0.9521 | 0.9267 | 1.0000 |
| 600 | 0.8276 | -0.0035 | 0.7337 | 0.6383 | 1.0000 |

Table 8: Ablation study: results for different numbers of clients on digits.

**Effects of different FL algorithms** In Fig. 9, we show the standard accuracy, WSR and TAcc for our proposed DUW in two different FL settings: fedavg and fedprox (Li et al., 2020). According to the results, fedprox can achieve comparable WSR as fedavg and higher standard accuracy. TAcc for both FL algorithms remains to be 100%. Our proposed method is not sensitive to the FL framework, in which it is implanted.

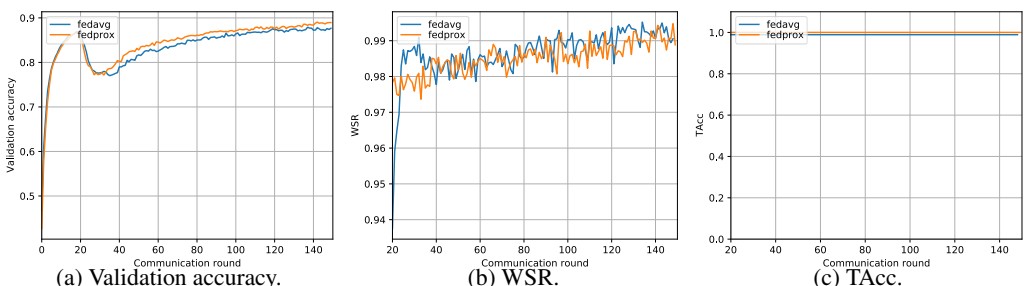

(a) Validation accuracy.     (b) WSR.     (c) TAcc.

Figure 9: Validation accuracy, WSR, and TAcc for fedavg and fedprox on digits.

### B.3 EXTENDED ROBUSTNESS STUDY

**Robustness against detection attack.** We take Neural Cleanse (Wang et al., 2019) as an example of the detection attack, which synthesizes the possible trigger to convert all benign images to all possible target classes in the classification task space. Then anomaly detection is conducted to detect if any trigger candidate is significantly smaller than other candidates. We follow the original setting in Wang et al. (2019), if the anomaly index is larger than 2, the model is watermarked. The smaller the value of the anomaly index, the harder the watermark to be washed out by Neural Cleanse. Local samples are used as benign images during detection. We compare the anomaly index for the non-watermarked model and watermarked model in Fig. 10. We observe that for all datasets, the anomaly index for the watermarked model is close to that of the non-watermarked model, and both of them are smaller than the threshold 2. The observation implies that our watermarked model cannot be detected using neural cleanse. One possible reason is that Neural Cleanse relies on the assumption that the backdoor-based watermark shares the same task space with the original classification task, but due to the effectiveness of our decoder, our target label space of the watermark is different from the original classification task space. Therefore, by searching all possible target classes in the original task space, Neural Cleanse will not find the real target label of the trigger set introduced by our proposed DUW.

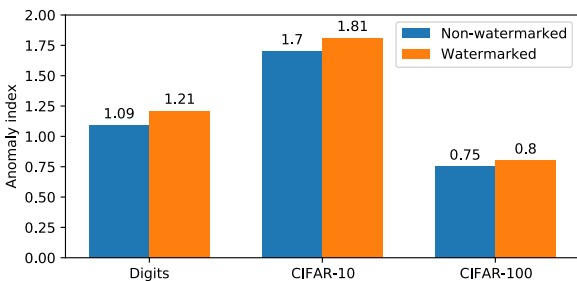

Figure 10: Anomaly index of watermarked model and non-watermarked model. If the anomaly index exceeds 2, the model will be detected as backdoor-based watermarked.

We further show the reversed trigger pattern generated by Neural Cleanse for non-watermarked (non-wm) and watermarked (wm) models in Fig. 11. The reversed trigger of our watermarked model shares a similar pattern as non-watermarked ones for all three benchmarks, and it does not look similar to our real trigger patterns (real ones can be referred to Fig. 6 residual). The trigger patterns for our trigger sets are sample-specific. Thus, it is hard to reverse engineer triggers when Neural Cleanse assumes a general trigger pattern for the entire trigger set. In summary, our proposed DUW is secured against this trigger-detection algorithm.

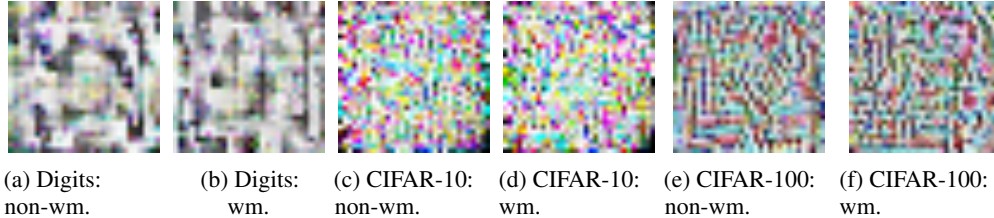

| (a) Digits: non-wm. | (b) Digits: wm. | (c) CIFAR-10: non-wm. | (d) CIFAR-10: wm. | (e) CIFAR-100: non-wm. | (f) CIFAR-100: wm. |

Figure 11: Reversed trigger patterns generated by Neural Cleanse for non-watermarked (non-wm) and watermarked (wm) models on three benchmarks.

## B.4 HYBRID WATERMARK

If the server wants to verify the model ownership quickly in a black-box way before tracking the infringers, our DUW can also be combined with existing global unified watermarks. We design a simple hybrid watermark in this section as an example. We pick one of the trigger sets we generated for the clients as the trigger set for the unified watermark injection, and the target label is assigned as 0 which belongs to the original label set of the training data. We use this trigger set to fine-tune the entire global model for 10 steps before injecting our proposed DUW. Note that no decoder is used for the unified watermark, and the unified watermarks can also be replaced with other existing works. The results on Digits are shown in Table 9. For this table, we can observe that the unified watermark is injected successfully in the presence of our DUW, with a $98.82\%$ WSR. Besides, the effectiveness of our DUW is also not affected, since the WSR of DUW only decreases by $0.72\%$, and TAcc remains $100\%$. The model utility is also not affected, since the standard accuracy remains high.

| Method | Acc | $\Delta$Acc | WSR | WSR_Gap | TAcc | Unified WSR |
|---|---|---|---|---|---|---|
| w/o unified watermark | 0.8855 | 0.0234 | 0.9909 | 0.9895 | 1.0000 | / |
| w/ unified watermark | 0.8886 | 0.0203 | 0.9837 | 0.9701 | 1.0000 | 0.9882 |

Table 9: Results for hybrid watermark.