# OpenReview forum: "Who Leaked the Model? Tracking IP Infringers in Accountable Federated Learning"
_ICLR.cc/2024/Conference — Submitted to ICLR 2024_

### Official Review · Reviewer_MDWZ · 2023-10-14

**Soundness:** 3 good
**Presentation:** 3 good
**Contribution:** 3 good
**Rating:** 8
**Confidence:** 3

**Summary:**

The authors present a watermarking scheme for FL that allows to clearly and reliably identify which client leaked the global model. The so-called DUW scheme follows a backdoor-based approach where, after each aggregation round, the global model is backdoored to produce a client-specific target label when using a special decoder instead of the classification head. The authors also propose an optimization that aims to preserve the utility of the actual model and works by limiting the distance between original and backdoored model. The evaluation on standard image classification tasks shows that the proposed method is incredibly reliable, robust against several watermarking removal methods, and still preserves a high accuracy compared to training without watermarking.

**Strengths:**

The paper is well written and accessible even for non-experts. Prior and related works are clearly described and important research gaps identified. The resulting scheme seems to be a practically viable solution without any obvious drawbacks that fulfils all desired properties. The evaluation is extensive and all questions I had in mind were answered with meaningful experiments, e.g., the robustness is properly checked against multiple watermarking removal approaches.

**Weaknesses:**

I cannot find serious weaknesses in this paper. A few suggestions to improve the presentations are made below.

The discussion of related work primarily mentions FedTracker as relevant prior work. However, there also exist further works such as Merkle-Sign by Li et al. (arXiv:2105.03167 / ICMEW'22) and FedCIP by Liang and Wang (arXiv:2306.01356).

The generation of the trigger sets based on the pre-trained encoder of Li et al is not really explained. It would be great to get some more details how the encoding of client keys into the dataset works.

Algorithm 1, instead of simply referring to Equations 6 and 1, should make it more explicit where some of the defined values such as the set D_T are being used.

Instead of providing only the final benchmark results after all rounds in Table 1, plots showing the evolvement over rounds would be interesting.

**Questions:**

- How does DUW compare to the above mentioned works?

---

> ### Author Response · Authors · 2023-11-20
> **Thanks for your helpful comments and suggestions**
>
> Thank you very much for the reviewer's positive comment and constructive suggestions! We address your concerns as follows:
> 1) **The discussion of related work primarily mentions FedTracker as relevant prior work. However, there also exist further works such as Merkle-Sign by Li et al. (arXiv:2105.03167 / ICMEW'22) and FedCIP by Liang and Wang (arXiv:2306.01356).**
>
> Thank you very much for your comment! Both Merkle-Sign and FedCIP are client-side watermarking, which is different from our server-side watermarking. Client-side watermarking is used to claim the co-ownership of the model, yet we argue that client-side watermarking has some limitations, which makes it not applicable for IP tracking. For client-side watermarking, if one of the clients is the infringer to illegally distribute the model, the infringer will not reveal their own identity during the model verification process in order to avoid legal responsibility. Even if the ownership of the model can be claimed by their co-author, the real infringers cannot be tracked, since they remain anonymous. Using our server-side watermark, there is no such concern, the server can easily track the malicious client among all the clients. We add the discussions of these approaches in A. Discussions “Client-side watermarking VS server-side watermarking” in the supplementary.
>
> 2) **The generation of the trigger sets based on the pre-trained encoder of Li et al is not really explained. It would be great to get some more details how the encoding of client keys into the dataset works.**
>
> The pre-trained encoder has the same architecture as Li et al. During the pre-training of the encoder, we encode an attacker-specified string into benign images through an encoder-decoder network (The decoder here will not be used in our paper). The mapping from the string to the target label will be generated when DNNs are trained on the poisoned dataset. The generated triggers are invisible additive noises containing a representative string of the target label.
>
> 3) **Algorithm 1, instead of simply referring to Equations 6 and 1, should make it more explicit where some of the defined values such as the set D_T are being used.**
>
> Thank very much for your suggestion. In the revised version of Algorithm 1, we add how trigger sets, and local training sets are used.
>
> 4) **Instead of providing only the final benchmark results after all rounds in Table 1, plots showing the evolvement over rounds would be interesting.**
>
> Thank you very much for your suggestion. The Acc and WSR curve with respect to the communication rounds can be found in Figure 4b. Acc will decrease once the watermark is injected, but will quickly resume after a few rounds. WSR will increase to a high value, and then remain stable.
>
> 5) **How does DUW compare to the above mentioned works?**
>
> Thank you very much for your comment. Since we have answered in question 1 that the mentioned works do not address the same problem setting as our paper, we discuss the different problem setting between the mentioned works and our DUW in in A. Discussions “Client-side watermarking VS server-side watermarking” in the supplementary.

---

> > ### Comment · Reviewer_MDWZ · 2023-11-22
> > **Thanks for the rebuttal**
> >
> > Thanks for clarifying my concerns and answering my question!

---

> > > ### Author Response · Authors · 2023-11-22
> > >
> > > Thank you very much for carefully reading our response! We are glad our response has addressed your concerns.

---

### Official Review · Reviewer_qn5H · 2023-10-31

**Soundness:** 2 fair
**Presentation:** 2 fair
**Contribution:** 2 fair
**Rating:** 3
**Confidence:** 3

**Summary:**

The goal of this paper is to create a watermarking schema for federated learning. The watermark should not only be able to help us identify the stolen model but also indicate which client leaked the model. The essential requirements for the watermark are: (1) accurate IP tracking - identify the client who leaked the collaboratively trained model, (2) provide the confident output of the ownership verification and the identification of the client who leaked the model, (3) the injected watermark should not lower the quality of the model, (4) the watermark should not be easy to remove, e.g., by fine-tuning. The main method assumes that the central server watermarks the shared model before sending it to the clients by assigning to each client a separate dataset. The method is expensive on the server side when we consider millions or more clients.

**Strengths:**

1. The problem is valid. We want to find out who leaked the collaboratively trained model.
2. The usage of the encoder-decoder from Li et al. (2021b) to generate unique trigger sets for each client is an interesting solution to lower the burden put on the server (step 1 on page 4).

**Weaknesses:**

1.  Verification: "To achieve this goal, we first use our decoder D to replace the classifier h_s in the suspect model $M_s$, then the suspect model can be restructured as $M_s = (f_s , D)$" - this is the biggest flaw in the paper. It was claimed on page 3 that: "Shao et al. (2022) proposed a parameter-based watermarking method for FL called FedTracker. It inserts a unique parameter-based watermark into the models of each client to verify the ownership. However, all parameter-based watermarking requires an **inspection of the parameters of the suspect models, which is not applicable enough for many re-sale models**". However, this method also requires access to the parameters of the suspect model to replace h_s with D during the verification process. If this is not the case, then the authors should explain clearly how to decode the keys from the suspect model.
1. The method is impractical for FL across devices where we can deal with millions or more clients. It assumes that "During each communication round, the server watermarks the aggregated global model using the client-wise trigger sets before dispatching the model." and it aims at a "traceable IP verification for accountable FL that can accurately identify the infringers among a **scalable** number of clients". It was remarked that the early training rounds can be skipped but only the first 20 out of a total 300 for CIFAR10 (beginning of page 7). Furthermore: " in order to avoid this pitfall, we have to ensure the uniqueness of both the triggers and target labels between different clients".  Overall, this method is excessively expensive for the server!
2. There can be a false positive if the client has some additional data from the data used for the watermarking and the potential watermark collisions between different clients.
3. The watermark is broken at the very core - if we test the ownership by sending the trigger sets produced for each client, then this requires a lot of queries.
4. If there are many Sybils or colluding parties, they could use the same encoder from Li et al. (2021b) to embed the watermark. The method would detect the same watermark for many models, which would make the verification of the client that leaked the model impossible since it is not a single client that leaks the shared model.
5. The authors did not release the source code so it is not possible to check the details of the method.


Minor comments:
- On page 5, Subsection 3.4 $M_s$ is used for both $(f_s , h_s )$ and $(f_s, D)$.
- Figure 1 is too complex and difficult to understand here - what is the decoder?
- page 2 - method description - what is the pre-trained encoder?
- at the end of page 2: "our work can be summarized in four folds" - but you have only 3 contributions enumerated
- from the initial description on page 2 - it should be already explained how the watermark despite being produced per client by the server affects the aggregation of the model updates/parameters
- "distributed learning framework that enables massive and remote clients"  page 3 - what are the massive clients?
- page 2 or 3 - I would like to learn how big have to be the separate dataset/trigger sets $D_T$ for each client. How much additional data does the server have to prepare? How much different the datasets have to be for each client?
- "the server will inject a unique watermark for each client" - again, this exerts the whole work on the server - which is too big of an overhead.
-

**Questions:**

1. What is the exact setup for CIFAR10 and CIFAR100? How many clients? How many data points per client? What exact models / encoders / decoders are used?
2. How is the decoder used for the verification process?
3. Do you need to replace the classifier with the decoder for the verification?
4. Would you improve the notation? On page 5, Subsection 3.4 $M_s$ is used for both $(f_s , h_s )$ and $(f_s, D)$.
5. Would you improve Figure 1? It it too complex but still does not explain how the method works. How does the decoder work?
6. Would you add the ablation study for the size of the key pool?
7. Why does fine-tuning increase the accuracy in Table 2 for Digits and CIFAR10? Why does accuracy drop for CIFAR100?

---

> ### Author Response · Authors · 2023-11-20
> **Thanks for your helpful comments and suggestions - Part 1**
>
> We are glad that the reviewer found our method interesting. We thank the reviewer for the constructive comments and suggestions, which we address below:
> 1) **Verification: "To achieve this goal, we first use our decoder D to replace the classifier h_s in the suspect model " - this is the biggest flaw in the paper. It was claimed on page 3 that: "Shao et al. (2022) proposed a parameter-based watermarking method for FL called FedTracker. It inserts a unique parameter-based watermark into the models of each client to verify the ownership. However, all parameter-based watermarking requires an inspection of the parameters of the suspect models, which is not applicable enough for many re-sale models". However, this method also requires access to the parameters of the suspect model to replace h_s with D during the verification process. If this is not the case, then the authors should explain clearly how to decode the keys from the suspect model.**
>
> Thank you very much for your comment. We argue that our approach is more applicable than parameter-based methods, because many of the IP infringers may illegally distribute their pre-trained feature extractor (encoder) $f$, and the classifier $h$ is just a linear layer without much pre-trained information. In this scenario, our method does not require to have access to any of the parameter distributions of the suspect model.
>
> In another scenario, if the IP infringers distribute the entire model, we also have an alternative solution denoted as a Hybrid watermark, which is shown in B.4 in the supplementary. Our DUW can also be combined with existing global unified watermarks. The unified watermark is to detect the mode leakage first in a black-box way. If we can claim ownership of the model, it is reasonable to ask the third party to provide the model to track the source of the IP leakage using our proposed DUW.
>
> 2) **The method is impractical for FL across devices where we can deal with millions or more clients. It assumes that "During each communication round, the server watermarks the aggregated global model using the client-wise trigger sets before dispatching the model." and it aims at a "traceable IP verification for accountable FL that can accurately identify the infringers among a scalable number of clients". It was remarked that the early training rounds can be skipped but only the first 20 out of a total 300 for CIFAR10 (beginning of page 7). Furthermore:  in order to avoid this pitfall, we have to ensure the uniqueness of both the triggers and target labels between different clients.**
>
> Thank you very much for your comment. In this paper, instead of a cross-device setting, we target a cross-silo setting of FL, which is also a widely adopted and practical setting [1-3], including FL IP protection setting [4]. In the cross-silo setting, each client is an institute, like a hospital or a bank. The number of clients is usually set as hundreds.  Our work aims to prototype and focus on an IP tracking technique and leave the scalability for the future. To address the confusion, in the updated version of the paper, we have add the corresponding clarification at the beginning of section 2.
>
> We also add ablation studies with more clients in Table 8 in the supplementary. According to the results, even with 600 clients, the WSR is still over 73%, and the TAcc remains 100%. With more clients participating in FL, we can still track the malicious client correctly with high confidence. Note that the largest number of clients in the table is not the upper limit of the capacity of our proposed DUW.
>
> [1] Bagdasaryan E, Veit A, Hua Y, et al. How to backdoor federated learning[C]//International conference on artificial intelligence and statistics. PMLR, 2020: 2938-2948.
>
> [2] T Dinh C, Tran N, Nguyen J. Personalized federated learning with moreau envelopes[J]. Advances in Neural Information Processing Systems, 2020.
>
> [3] Zhu Z, Hong J, Zhou J. Data-free knowledge distillation for heterogeneous federated learning[C]//ICML2021.
>
> [4] Tekgul B G A, Xia Y, Marchal S, et al. Waffle: Watermarking in federated learning[C]//2021 40th International Symposium on Reliable Distributed Systems (SRDS). IEEE, 2021.
>
> 3) **"the server will inject a unique watermark for each client" - this exerts the whole work on the server, which is too big of an overhead, excessively expensive for the server!**
>
> The additional computation for the server is decided by the number of watermark injection steps T_w. We found that WSR could reach 99\% just within $T_w=10$ steps. The server can embed the watermark parallelly for all the clients. Since the watermarked model for each client is independent and has no sequence relationship with each other, there is no need to serialize it. Not all clients will be active for each communication round, and the client-unique watermark will only be injected to the active clients. Thus, the computation cost will not be so expensive for the server.

---

> ### Author Response · Authors · 2023-11-20
> **Thanks for your helpful comments and suggestions - Part 2**
>
> 4) **There can be a false positive if the client has some additional data from the data used for the watermarking and the potential watermark collisions between different clients.**
>
> Thank you very much for your comment. Our experiment results in Table 5 and Figure 4b show that even if we use random noise or randomly generated jigsaw for watermark injection we can get 100% TAcc, suggesting no collision of clients. If random noise or jigsaw is applied for watermark injection, the clients will not get the data that shares the same distribution as the watermark data.
>
> 5) **The watermark is broken at the very core - if we test the ownership by sending the trigger sets produced for each client, then this requires a lot of queries.**
>
>  Thank you very much for your comment. We agree that this may pose an additional challenge to individual watermarks, and we have proposed a solution that combines client watermark and a global watermark, denoted as a Hybrid watermark, which is shown in B.4 in the supplementary. Our DUW can also be combined with existing global unified watermarks. The unified watermark is to detect the mode leakage first in a black-box way. Only if we can claim ownership of the model using the global unified watermark, we will try to track the identity of the infringers by adopting  DUW. Thus, we can avoid sending so many queries to all the suspect models. We will only send client-wise trigger sets to the models that we have already claimed ownership of to find the anonymous infringer.
>
> 6) **If there are many Sybils or colluding parties, they could use the same encoder from Li et al. (2021b) to embed the watermark. The method would detect the same watermark for many models, which would make the verification of the client that leaked the model impossible since it is not a single client that leaks the shared model.**
>
> We want to argue that the watermark injection is not conducted on the client side, it is conducted by the server, which ensures a unique watermark is injected for each client before distributing the watermarked model to the clients. Thus, it is not possible to detect the same watermark for different clients’ models.
>
> 7) **The authors did not release the source code so it is not possible to check the details of the method.**
>
> We have revised the paper to provide more details for reproducibility. We promise to release codes upon acceptance.
>
> 8) **On page 5, Subsection 3.4 M_s is used for both $(f_s,ℎ_s)$ and $(f_s,D)$.**
>
> The suspect model is originally constructed of $(f_s,h_s)$, and then we reconstruct it as $(f_s,D)$. Thus, we use the same $M_s$.
>
> 9) **Figure 1 is too complex and difficult to understand here - what is the decoder?**
>
> We introduce the architecture of the decoder D in step2 in section 3.2. The decoder $D$ only has one linear layer, whose input dimension is the same as the input dimension of classifier head $h$, and its output dimension is the length of the key.  Figure 1 of the revised manuscript is also revised for easier understanding.
>
> 10) **page 2 - method description - what is the pre-trained encoder?**
>
> The pre-trained encoder has the same architecture as [1]. During the pre-training of the encoder, we encode an attacker-specified string into benign images through an encoder-decoder network (The decoder here will not be used in our paper). The mapping from the string to the target label will be generated when DNNs are trained on the poisoned dataset. The generated triggers are invisible additive noises containing a representative string of the target label.
>
> [1]  Y. Li, Y. Li, B. Wu, L. Li, R. He, and S. Lyu. Invisible backdoor attack with sample-specific triggers. In Proceedings of the IEEE/CVF International Conference on Computer Vision, pages16463–16472, 2021.
>
> 11) **at the end of page 2: "our work can be summarized in four folds" - but you have only 3 contributions enumerated**
>
> Thanks for the catch, and we have corrected this error.
>
> 12) **from the initial description on page 2 - it should be already explained how the watermark despite being produced per client by the server affects the aggregation of the model updates/parameters**
>
> Thank you very much for your comment. We investigate how the watermarking will affect model aggregation in Figure 4 (b). Compared with standard training without watermarking (purple line), the standard accuracy has a faster recovery of the standard accuracy after the accuracy drop at the watermark injection round. The final standard accuracy degradations of watermarked model are only 2.34\%, 0.03\% and 0.63\% for digits, cifar10, and cifar100, respectively.
>
> 13) **"distributed learning framework that enables massive and remote clients" page 3 - what are the massive clients?**
>
> In our cross-silo FL setting, massive clients indicate a large number of participating institutes. The participating institutes (clients) are typically large in number and have slow or unstable internet connections.

---

> ### Author Response · Authors · 2023-11-20
> **Thanks for your helpful comments and suggestions - Part 3**
>
> 14) **page 2 or 3 - I would like to learn how big have to be the separate dataset/trigger sets for each client. How much additional data does the server have to prepare? How much different the datasets have to be for each client?**
>
> The size of the trigger set is 500 as stated in section 4 datasets. A client-unique key is encoded into the trigger sets using the pre-trained encoder. The difference between the trigger sets can be demonstrated by the zero watermark collision (100\% tracking accuracy) between clients.
>
> 15) **What is the exact setup for CIFAR10 and CIFAR100? How many clients? How many data points per client? What exact models / encoders / decoders are used?**
>
> The setup of CIFAR10 and CIFAR100 including the number of clients and how the data is partitioned is stated in Datasets in section 4. The architecture of the model is stated in the training setup in section 4. The architecture of the encoder is the same as [1]. We introduce the architecture of the decoder D in step2 in section 3.2. The decoder $D$ only has one linear layer, whose input dimension is the same as the input dimension of classifier head $h$, and its output dimension is the length of the key.
>
> [1]  Y. Li, Y. Li, B. Wu, L. Li, R. He, and S. Lyu. Invisible backdoor attack with sample-specific triggers. In Proceedings of the IEEE/CVF International Conference on Computer Vision, pages16463–16472, 2021.
>
> 16) **How is the decoder used for the verification process? Do you need to replace the classifier with the decoder for the verification?**
>
> During verification, to verify if the triggers can be decoded as the corresponding keys, we first use our decoder $D$ to replace the classifier $h_s$ in the suspect model $M_s$, and then test on the verification dataset. This part is stated in section 3.4, and ‘Server: verification’ in Figure 1.
>
> 17) **Would you improve Figure 1? It it too complex but still does not explain how the method works. How does the decoder work?**
>
> In Figure 1 of the revised manuscript, we added one simple description of the decoder and simplified the framework for easier understanding. The decoder is used to replace the classifier head as shown in Figure 1. We introduce how the decoder works in detail in section 4.2 Step2. “The decoder $D$ only has one linear layer, whose input dimension is the same as the input dimension of $h$, and its output dimension is the length of the key. To avoid watermark collision between clients induced by the target label, we make the decoder weights orthogonal with each other during the random initialization so that the watermark injection tasks for each client can be independent . The weights of the decoder are frozen once initialized to preserve the independence of different tasks for different clients during watermark injection.”
>
> 18) **Would you add the ablation study for the size of the key pool?**
>
> The key is not randomly selected, we design a unique key corresponding to each client ID as a one-hot binary string to differentiate clients. For instance, for the $k$-th client, the $k$-th entry of the key string $s_k$ is $1$, and the other entries are $0$. Thus, changing the size of the key pool will not affect the results.
>
> 19) **Why does fine-tuning increase the accuracy in Table 2 for Digits and CIFAR10? Why does accuracy drop for CIFAR100?**
>
> The local dataset is used for fine-tuning. Due to the limited dataset size and non-iid distribution, the standard accuracy is likely to drop. For Digits, the model architecture is CNN, which is a much simpler architecture compared with ResNet for CIFAR-10 and CIFAR-100, the learning of Digits is also simpler, thus, thus, additional fine-tuning is more likely to improve the standard accuracy.

---

> ### Author Response · Authors · 2023-11-22
> **A kind reminder to reviewer qn5H**
>
> Dear Reviewer qn5H,
>
> Thank you for your time to review our paper and leave valuable comments and suggestions. As this is the last day of the discussion, we are wondering whether you have had a chance to read our response to your questions. We will be glad to provide more explanations and answer more questions if you have any.
>
> Authors

---

### Official Review · Reviewer_Khiu · 2023-11-01

**Soundness:** 3 good
**Presentation:** 3 good
**Contribution:** 2 fair
**Rating:** 5
**Confidence:** 5

**Summary:**

This paper proposes a novel method to inject backdoor-based watermark to track IP infringers in the FL setting. Using an encoder-decoder framework, this paper encodes the client unique IDs into the federated model. Experimental results demonstrate the effectiveness of the approach.

**Strengths:**

1. This work addresses an important and timely problem, which is to not only inject watermarks to protect model IPs but also track the IP leakages in FL settings.
2. The paper is well-written and easy to follow in general.

3. Experimental evaluations are comprehensive, covering a broad number of aspects and ablation studies.

**Weaknesses:**

1. The idea of using encoder-decoder framework to embed an identifiable string such as Labels is not new[1], therefore using encoder-decoder to identify client IDs, which is the main idea of this work, appears to be an straightforward extension and not very challenging. Experimental results in Table 1 also show perfect track score and high WSR_gap for all datasets, which seems to indicate that the underlining problem is not very challenging.  It is suggested that the authors provide more discussions on the unique challenges on identifying clients as compared to other identification problems.

2. The proposed method is based on the assumption that the client set is known and therefore an ID string can be assigned. In reality client sets are dynamic, especially in cross-device FL settings. How will the proposed algorithm deal with dynamic increase or decrease of the client set? Also since the decoder's dimension is higher than the number of clients, will this create scalability problems when the number of clients grow very large (e.g. millions) ?

3. The experimental results do not compare with other baseline methods. Are there any other backdoor watermarking approaches that worth comparing with?


[1] Li et al, Invisible backdoor attack with sample-specific triggers. In Proceedings of the IEEE/CVF International Conference on
Computer Vision, 2021.

**Questions:**

In Eq 6 and algorithm 1, \theta_k^f appears from nowhere without clear explanations. I suppose it is the feature exactor of \theta_k, is it correct?

---

> ### Author Response · Authors · 2023-11-20
> **Thanks for your helpful comments and suggestions - Part 1**
>
> We are glad that the reviewer found our problem setting important and our experiments comprehensive. We thank the reviewer for the constructive comments and suggestions, which we address below:
> 1) **The idea of using encoder-decoder framework to embed an identifiable string such as Labels is not new, therefore using encoder-decoder to identify client IDs, which is the main idea of this work, appears to be an straight forward extension and not very challenging. Experimental results in Table 1 also show perfect track score and high WSR_gap for all datasets, which seems to indicate that the underlining problem is not very challenging. It is suggested that the authors provide more discussions on the unique challenges on identifying clients as compared to other identification problems.**
>
> Thank you very much for your comment. To outline the challenges of this paper, we introduce the pitfalls for more straightforward backdoor-based methods in section 3.1 in the paper, and Appendix B.1 in the supplementary. According to the results, traditional backdoor-based watermarking with arbitrary triggers added for each client will cause at least 87\% of the clients to have watermark collisions among 100 clients. The watermark collision is a severe problem that remains unsolved for existing methods. Our experiment results with 100\% tracking accuracy and a high WSR_gap verify the effectiveness of our proposed method to track the IP infringers not only accurately but also confidently without collisions.
>
> 2) **The proposed method is based on the assumption that the client set is known and therefore an ID string can be assigned. In reality client sets are dynamic, especially in cross-device FL settings. How will the proposed algorithm deal with dynamic increase or decrease of the client set? Also since the decoder's dimension is higher than the number of clients, will this create scalability problems when the number of clients grow very large?**
>
> Thank you very much for your comment. In this paper, instead of a cross-device setting, we adopt a cross-silo setting of FL, which is also a widely adopted and practical setting [1-3], including FL IP protection setting [4]. In the cross-silo setting, each client is an institute, like a hospital or a bank. The number of clients is usually set as hundreds. To address the confusion, in the updated version of the paper, we have add the corresponding clarification at the beginning of section 2.
>
> The decrease in the client number will not affect the effectiveness of the method. As for the increase in client number, we add ablation studies with more clients in Table 8 in the supplementary. According to the results, even with 600 clients, the WSR is still over 73\%, and the TAcc remains 100\%. With more clients participating in FL, we can still track the malicious client correctly with high confidence. Note that the largest number of clients in the table is not the upper limit of the capacity of our proposed DUW. We believe that protecting high-performance models is more valuable in IP protection. Yet, scaling the client number to millions with limited data may cause poor performance. Our work aims to prototype and focus on an IP tracking technique and leave the scalability for the future.
>
> [1] Bagdasaryan E, Veit A, Hua Y, et al. How to backdoor federated learning[C]//International conference on artificial intelligence and statistics. PMLR, 2020.
>
> [2] T Dinh C, Tran N, Nguyen J. Personalized federated learning with moreau envelopes[J]. Advances in Neural Information Processing Systems, 2020.
>
> [3] Zhu Z, Hong J, Zhou J. Data-free knowledge distillation for heterogeneous federated learning[C]//ICML 2021.
>
> [4] Tekgul B G A, Xia Y, Marchal S, et al. Waffle: Watermarking in federated learning[C]//2021 40th International Symposium on Reliable Distributed Systems (SRDS). IEEE, 2021.

---

> ### Author Response · Authors · 2023-11-20
> **Thanks for your helpful comments and suggestions - Part 2**
>
> 3) **The experimental results do not compare with other baseline methods. Are there any other backdoor watermarking approaches that worth comparing with?**
>
> Thank you very much for your comment. Since tracking IP infringers with backdoor-based watermarking remains a new and open problem, and there is no existing research discussing a feasible solution. Therefore, we compare our proposed DUW with baselines designed by ourselves. Due to the reason that if all the clients share the same trigger, watermark collision will definitely happen, we design different triggers for different clients. Specifically, we use traditional backdoor-based watermarking by adding arbitrary badnet triggers using random noise or 0-1 coding trigger for each client. The target label for each client is set as (client\_{ID} \% class\_number). According to the results, traditional backdoor-based watermarks can only achieve a tracking accuracy lower than 13\% (it will even be lower with the increase of the communication rounds), which is much lower than the 100\% tracking accuracy we have achieved. Note that, the rate of clients with watermark collisions can be calculated as 1-TAcc. The results are shown in in section 3.1 in the paper, and Appendix B.1 in the supplementary.
>
> 4) **In Eq 6 and algorithm 1, \theta_k^f appears from nowhere without clear explanations. I suppose it is the feature exactor of \theta_k, is it correct?**
>
> Thank you very much for your comment. ${\theta_k=(\theta_k^f, \theta_k^h)}$ is the parameter which will be broadcast for client $k$  by the server, where $\theta_k^f$ is the feature extractor, and $\theta_k^h$ is the classifier head. In the revised manuscript, we add this definition before Eq. (4), where $\theta_k^f$ first appears.

---

> ### Author Response · Authors · 2023-11-22
> **A kind reminder to reviewer Khiu**
>
> Dear Reviewer Khiu,
>
> Thank you for your time to review our paper and leave valuable comments and suggestions. As this is the last day of the discussion, we are wondering whether you have had a chance to read our response to your questions. We will be glad to provide more explanations and answer more questions if you have any.
>
> Authors

---

> > ### Comment · Reviewer_Khiu · 2023-11-23
> >
> > I have read the responses and would like to keep my ratings.

---

### Meta-Review · Area_Chair_fjGY · 2023-12-07

**Metareview:**

This submission describes an approach to verification of model ownership in a decentralized, federated learning setup. The model watermark encoding strategy of Li et al., is used to encode ownership information into each checkpoint before it is sent to users. This approach is tested in a range of experiments in the cross-silo setting.

Reviewers commended the timeliness and the importance of the problem considered in this work. However, the reviews also brought up a number of credible concerns, such as the limited practical improvement over parameter-access-based federated watermarking, significant computational effort and noticeable drop in model accuracy. We explicitely note that reviewers have carefully read the authors' response and I have read the AC response, and we did not find the raised concerns to be solved.

**Justification For Why Not Higher Score:**

Concerns brought up during the review process as described above.

**Justification For Why Not Lower Score:**

N/A

---

### Decision · Program_Chairs · 2024-01-16

Reject